# Evaluation of the Insertion Depth of the iStent Inject W and Its Association with Surgical Outcomes Using Automated Gonioscopy

**DOI:** 10.3390/jcm14217547

**Published:** 2025-10-24

**Authors:** Yuki Takagi, Ryo Asano, Kana Yamashita, Ayumi Miwa, Yukihiro Sakai, Sho Yokoyama, Kei Ichikawa, Kazuo Ichikawa

**Affiliations:** 1Department of Ophthalmology, Japan Community Healthcare Organization Kanitounou Hospital, Kani 509-0206, Gifu, Japan; 2Chukyo Eye Clinic, Nagoya 456-0032, Aichi, Japan; yamashita.kanna@chukyo-eye.or.jp (K.Y.); miwa.ayumi@chukyo-eye.or.jp (A.M.); sakai@chukyo-eyeclinic.jp (Y.S.); yokoyama@chukyogroup.jp (S.Y.); kei@chukyogroup.jp (K.I.); ichikawa@chukyogroup.jp (K.I.); 3Asano Eye Clinic, Nagoya 455-0801, Aichi, Japan; asano@chukyogroup.jp

**Keywords:** glaucoma, iStent, gonioscopy, automated gonioscopy

## Abstract

**Background/Objectives**: The iStent is a trabecular micro-bypass device used in glaucoma management. Its depth has been evaluated using anterior segment optical coherence tomography; however, no reports have assessed it using gonioscopy. We evaluated the depth of the iStent inject W using the GS-1 automated gonioscope (GS-1) and determined its association with outcomes. **Methods:** Patients who underwent iStent inject W implantation were followed for 1 year. Angle photographs were obtained using a GS-1. The depth was graded from the GS-1 images as Grades 1 (anterior edge of the flange protruding into the anterior chamber) and 2 (anterior edge embedded within the trabecular meshwork). Based on the two implants’ depths, the eyes were categorized into Grades 1 + 1, 2 + 2, and 1 + 2. The percentages of intraocular pressure (IOP) reduction at 12 months were compared among the groups using the Kruskal–Wallis test. **Results**: Data from 31 patients (41 eyes and 82 stents) were analyzed. The depth distribution for Grades 1 and 2 was 51 (62.2%) and 31 (37.8%) stents, respectively. In the Grade 1 + 1, 2 + 2, and 1 + 2 groups, the number of eyes was 18, 8, and 15, respectively. IOP reductions at 12 months were 12.81 ± 20.27%, 22.85 ± 15.30%, and 16.04 ± 20.41% for the Grade 1 + 1, 2 + 2, and 1 + 2 groups, respectively, with no significant differences (*p* = 0.493). **Conclusions:** The impact of insertion depth on outcomes appears to be minimal in eyes where the iStent inject W was visualized using the GS-1. Further studies with larger cohorts should clarify the effects of insertion depth and its interplay with position.

## 1. Introduction

Intraocular pressure (IOP) reduction is the only established therapy for glaucoma [1]. Conventional surgical techniques for glaucoma are broadly categorized into filtration procedures (such as trabeculectomy) and trabeculotomy, which targets the trabecular outflow pathway. Although trabeculectomy achieves substantial IOP reduction, it requires conjunctival dissection and iridectomy, making it relatively invasive. Traditional trabeculotomy was also performed ab externo, similarly necessitating a conjunctival incision and potentially compromising subsequent trabeculectomy in some cases. Recent advancements have been made in minimally invasive glaucoma surgery. Several studies have demonstrated the safety and efficacy of trabecular outflow reconstruction procedures using devices such as the Trabectome [2,3,4,5,6] and implants inserted into the trabecular meshwork (TM), including the iStent series (Glaukos Corporation, Laguna Hills, CA, USA) [7,8,9,10] and Hydrus [11,12]. However, these devices are usually malpositioned relative to the TM (in terms of height, angle, or depth), and their postoperative positions cannot be confirmed via gonioscopy in some cases.

The GS-1 automated gonioscope (GS-1, NIDEK) can automatically capture color photographs of the entire angle across 16 directions. It enables rapid and relatively noninvasive examination [13,14,15] and facilitates longitudinal comparisons of angle structures. The GS-1 has also been used to evaluate the formation of peripheral anterior synechiae (PAS) after trabecular outflow reconstruction [16]. Zimmerman et al. used GS-1 to analyze the effect of iStent inject^®^ (Glaukos Corporation, Laguna Hills, CA, USA) at the implantation site on treatment efficacy [17]. However, they reported that the eyes with Grade 4 placement (insertion into the ciliary body band) showed a reduction in IOP comparable to that observed in eyes with Grade 2 or 3 placement near the TM.

Gillmann et al. reported the positions of the implanted iStent inject devices based on anterior segment optical coherence tomography (AS-OCT) and their outcomes according to their depths in relation to the TM [18,19]. They reported that greater implant protrusion into the anterior chamber and increased gonioscopic visibility of the device were associated with better surgical outcomes. However, to the best of our knowledge, no studies have reported the insertion depth of the iStent inject W based on AS-OCT or gonioscopic examination or its association with surgical outcomes.

Therefore, the present study aimed to use GS-1 to determine the insertion depth of the iStent inject W and assess its impact on treatment efficacy.

## 2. Materials and Methods

This was a retrospective study, with data reviewed for research purposes on 4 March 2025. It was conducted in accordance with the Declaration of Helsinki and was approved by the Ethics Committee of Chukyo Eye Clinic on 18 July 2024 (approval number: 20240227075). An opt-out approach was adopted for obtaining informed consent owing to the study’s retrospective nature. In this study, the eyes were included if they met the following criteria: underwent combined cataract surgery and iStent inject W implantation, had intraoperatively confirmed secure positions of the two devices within the nasal–inferior quadrants, completed 12 months of follow-up, and had identifiable implantation sites on GS-1 images obtained at 6 or 12 months postoperatively.

### 2.1. iStent Inject W and Follow-Up

All eyes underwent combined cataract surgery and iStent inject W implantation. A clear temporal corneal incision was made for phacoemulsification and intraocular lens implantation. Next, two iStent inject W implants were inserted into the nasal–inferior quadrants at intervals of 60–90° after the anterior chamber was filled with an ophthalmic viscosurgical device. Postoperative topical medications, including levofloxacin hydrate, bromfenac sodium hydrate, and betamethasone sodium phosphate, were administered for approximately 1 month. Anti-glaucoma medications were adjusted postoperatively based on the treating physician’s judgment. The outcomes included the IOP measured using Goldmann applanation tonometry and medication scores at baseline and at 1, 3, 6, 9, and 12 months. Fixed-combination drops were scored 2 points, and oral acetazolamide was scored based on the number of tablets.

### 2.2. Automated Gonioscopy

Orthoptists obtained GS-1 images under bright illumination after topical anesthesia with oxybuprocaine hydrochloride. A single grader determined the implantation site and insertion depth based on the 16 directional images. The locations were graded according to the scheme of Zimmerman as follows: Grade 1, implant center over the nonpigmented to pigmented TM; Grade 2, within the pigmented TM; Grade 3, from the pigmented TM toward the scleral spur; and Grade 4, below the scleral spur within the ciliary body band. Insertion depth was graded based on the criteria described by Gillmann et al. [18,19], focusing on the relationship between the anterior chamber-facing superior edge of the flange and the TM. Grade 1 denoted the edge being external (anterior) to the TM, while Grade 2 denoted it being within the TM (Figure 1). We assessed whether the PAS abutted the flange based on the GS-1 images (Figure 2) and whether a TM cleft surrounded the implant (Figure 3). The same graders repeated the assessments on separate days to ensure reproducibility.

### 2.3. Statistical Analysis

For the primary endpoint, the eyes were categorized into three groups based on the iStent insertion depth as follows: Grades 1 + 1, 2 + 2, and 1 + 2. The percentage of IOP reduction from baseline to 12 months was compared among the three groups using the Kruskal–Wallis test with Bonferroni correction for multiple comparisons. We fitted generalized estimating equation (GEE) models with patient clustering and used the eye (right/left) as the intra-subject factor. An exchangeable working correlation was assumed. The dependent variable was the 12-month IOP reduction rate adjusted for the baseline IOP and medication score. Moreover, robust standard errors were calculated.

Secondary outcomes included IOPs and medication scores at baseline and 12 months. Eyes with only Grade 1 or 2 and 3 or 4 placements were assigned to groups 1 and 2, respectively. The 12-month IOP reduction rate and medication scores were compared between these two groups using the Mann–Whitney U test. We stratified the stents by insertion depth grade and used the chi-squared test to compare the frequencies of PAS abutting the flange and TM cleft surrounding the implant based on the GS-1 images. The eyes were further categorized into two groups based on the presence or absence of a TM cleft adjacent to the iStent at the eye level. Their IOPs and medication scores were compared at baseline and 12 months using the Mann–Whitney U test. Furthermore, κ statistics were used to assess the reproducibility of the GS-1–based grading of the implant location and depth. All statistical analyses were performed using IBM SPSS Statistics for Windows, version 29.0 (IBM Corp., Armonk, NY, USA). Statistical significance was set at *p* < 0.05.

A power analysis was performed using G*Power (version 3.1.9.7) for the primary endpoint, which is the comparison of the percentage of IOP reduction at 12 months postoperatively among the three depth-defined groups. Additionally, a one-way fixed-effects analysis of variance, as an approximation to the Kruskal–Wallis test, was applied. Statistical power was determined based on the pooled within-group standard deviation and corresponding effect size (Cohen’s f) calculated from our data. Furthermore, the minimum required sample size was estimated using the observed effect size, three groups, a significance level (α) of 0.05, and a statistical power of 0.8.

## 3. Results

A total of 33 patients (44 eyes) were initially identified. However, three eyes from two patients were excluded because they had three iStent inject W devices, as confirmed by GS-1 imaging and conventional gonioscopy findings. Ultimately, data from 31 patients (41 eyes and 82 stents) were included in the final analysis. Table 1 presents the baseline clinical characteristics. A summary of the raw data used for this study is shown in Appendix A. The insertion depths were classified as Grades 1 and 2 for 51 (62.2%) and 31 (37.8%) stents, respectively. In the Grade 1 + 1, 2 + 2, and 1 + 2 groups, the number of eyes was 18, 8, and 15, respectively. Table 1 presents the IOP, medication score, and IOP reduction for the overall cohort at baseline and 12 months. The IOP and medication score significantly decreased from 14.63 ± 4.43 to 12.05 ± 3.55 mmHg and 3.32 ± 1.59 to 2.44 ± 1.79, respectively (both *p* < 0.001). Table 2 presents the postoperative outcomes based on the insertion depth. The 12-month IOP reduction rates were 12.81 ± 20.27%, 22.85 ± 15.30%, and 16.04 ± 20.41% for the Grade 1 + 1, 2 + 2, and 1 + 2 groups, respectively, with no significant differences (*p* = 0.493). Similarly, the medication scores were 2.33 ± 1.41, 3.75 ± 1.83, and 1.87 ± 1.92 for the Grade 1 + 1, 2 + 2, and 1 + 2 groups, respectively, with no significant differences (*p* = 0.058). The 12-month IOP reduction differed across the three groups (12.39 ± 3.33, 9.50 ± 3.21, and 13.00 ± 3.55 mmHg for the Grade 1 + 1, 2 + 2, and 1 + 2 groups, respectively; *p* = 0.044). Post hoc testing revealed a significant difference between the Grade 1 + 2 and 2 + 2 groups (*p* = 0.047), and the overall effect of insertion depth was borderline (type III Wald χ^2^ = 5.88; df = 2; *p* = 0.061). The adjusted marginal mean 12-month IOP reduction rates were 0.136, 0.159, and 0.136 for the Grade 1 + 1, 1 + 2, and 2 + 2 groups, respectively, with no significant pairwise differences. Baseline IOP was the only significant predictor (*p* = 0.003).

Table 3 presents the implantation locations. Thirty-one and 10 eyes were allocated to Groups 1 and 2, respectively. The baseline IOPs and medication scores were 15.03 ± 4.91 mmHg and 3.23 ± 1.56 for Group 1 and 13.40 ± 2.12 mmHg and 3.60 ± 1.71 for Group 2, respectively, with no significant differences (*p* = 0.300 and 0.601). At 12 months postoperatively, the IOPs and medication scores were 11.74 ± 3.88 mmHg and 2.61 ± 1.87 for Group 1 and 13.00 ± 2.11 mmHg and 1.90 ± 1.45 for Group 2, respectively, with no significant differences (*p* = 0.099 and 0.286). The 12-month IOP reduction rates were 20.63 ± 17.54% and 1.44 ± 18.25% for Groups 1 and 2, respectively, with a significant difference (*p* = 0.006).

PAS was observed adjacent to seven stents in six eyes. The distribution of the stents based on insertion depth was six and one for Grades 1 and 2, respectively, with no significant differences (*p* = 0.180). TM clefts were identified around the implants in 10 stents across 10 eyes (Grade 1: 1; Grade 2: 9). The difference in cleft severity based on depth was significant (*p* < 0.001).

Patients were stratified based on the presence or absence of a TM cleft (Table 4). The baseline IOPs and medication scores were 14.71 ± 3.13 mmHg and 3.13 ± 1.48 for the no-cleft group and 14.40 ± 7.37 mmHg and 3.90 ± 1.85 for the cleft group, respectively, with no significant differences (*p* = 0.211 and 0.39, respectively). At 12 months postoperatively, the IOPs and medication scores were 12.58 ± 3.29 mmHg and 2.00 ± 1.57 for the no-cleft group and 10.40 ± 3.98 mmHg and 3.80 ± 1.81 for the cleft group, respectively, with significant differences (*p* = 0.04 and 0.0074). The 12-month IOP reduction rates were 13.75 ± 18.91% and 22.76 ± 20.22% for the no-cleft and cleft groups, respectively, with no significant difference (*p* = 0.207).

Stratification by PAS (Table 5) revealed baseline IOPs and medication scores of 14.26 ± 4.37 mmHg and 3.20 ± 1.64 for the PAS-negative group and 16.83 ± 4.49 mmHg and 4.00 ± 1.10 for the PAS-positive group, respectively, with no significant differences (*p* = 0.196 and 0.376). The IOPs and medication scores were 11.57 ± 3.53 mmHg and 2.31 ± 1.86 for the PAS-negative group and 14.83 ± 2.23 mmHg and 3.17 ± 1.17 for the PAS-positive group, respectively. The IOP significantly differed (*p* = 0.016), but not the medication scores (*p* = 0.284). Moreover, the 12-month IOP reduction rates for the PAS-negative and PAS-positive groups were 17.19 ± 19.65% and 8.69 ± 17.41%, respectively, with no significant difference (*p* = 0.284).

The intraobserver agreement was excellent for grading the insertion depth based on the GS-1 images (κ = 0.949; 95% confidence interval (CI): 0.878–1.000; *p* < 0.001). However, it was moderate for the implantation location (κ = 0.546; 95% CI: 0.387–0.705; *p* < 0.001).

Regarding the analysis of statistical power for the primary endpoint, based on the observed data (means: 12.81%, 22.85%, and 16.04%; standard deviations: 20.27%, 15.30%, and 20.41%; *n* = 18/8/15), the pooled within-group standard deviation was 19.50, while the corresponding effect size was calculated as Cohen’s f = 0.19. The post hoc power computed from these parameters was 0.16. Ultimately, the required sample size was calculated to be 288 eyes using the estimated effect size, three groups, a significance level (α) of 0.05, and a statistical power of 0.80.

## 4. Discussion

At 12 months postoperatively, the mean IOP and medication score decreased by 2.58 ± 3.27 mmHg (17.63%) and 0.88 ± 1.23 (27.33%), respectively. Previous studies have reported IOP reductions of approximately 17–24% [9,10,17], which are consistent with our findings. Notably, the baseline mean deviation of our cohort (−12.79 ± 8.34 dB) was more severe than that reported in the literature (−3.4 to −8.1 dB), indicating the inclusion of more advanced cases. These results suggest that a clinically meaningful reduction in IOP can be achieved for advanced glaucoma, and the extent of the reduction is comparable to that observed in milder disease.

The insertion depth and location of each iStent inject W were verified using the GS-1 images. In our cohort, the flange of 62.2% of the stents protruded into the anterior chamber (Grade 1). Previous reports [18,19] involving 19 patients (25 eyes) noted that the stent position could not be confirmed via gonioscopy in 12% of the eyes, and only 54.3% of the stents had a visible iStent inject flange protruding into the anterior chamber. Notably, these proportions are lower than those observed in our cohort. The earlier studies evaluated the iStent inject, which has a flange diameter of 240 μm, whereas the iStent inject W used in our series features a flange diameter of 360 μm (1.5 times larger). This size difference may reduce the risk of over-insertion of the iStent injection W and could explain the higher rate of anterior chamber protrusion observed in our cohort.

The eyes were categorized into three groups based on the iStent inject W insertion depths, and their 12-month IOP reduction rates were compared. No significant differences were observed among the groups. Insertion depth was not a significant predictor of IOP reduction in the GEE analysis. Previous studies [18,19] have reported better outcomes with greater anterior chamber protrusion of the iStent inject flange, which contrasts with our findings. Those series likely included cases involving implants inserted at depths that precluded their detection via gonioscopy. Gonioscopic visibility is a favorable prognostic factor. In our cohort, the implantation site and depth of all devices were verifiable on the GS-1 images. This suggests that extremely deep placements were rare, potentially explaining the absence of a depth–outcome association. However, the 12-month IOP was significantly different for the Grade 2 + 2 and 1 + 2 groups. The Grade 2 + 2 group had lower baseline IOP and higher postoperative medication scores than the other groups, although these differences were not statistically significant. These factors may have influenced the results. Therefore, future studies with larger sample sizes, better-matched baseline characteristics, and standardized postoperative medication protocols are warranted.

The IOPs and medication scores did not significantly differ between the two groups stratified by implantation location. However, group 1 had a significantly higher percentage of IOP reduction than group 2. This finding contrasts with that of a previous report [17], which found no significant difference in IOP reduction when stratifying eyes by implantation site. The study compared the eyes with Grade 4 and 2–3 placements. However, none of the cases in our cohort had Grade 4 placement alone. We compared the eyes with Grade 1–2 (within the TM) and 3 or 4 placements. These differences in the grouping criteria may account for the discrepant results. Future studies with larger cohorts and harmonized grouping definitions are needed to re-evaluate this question.

PAS was observed in seven stents across six eyes in the present study. While we anticipated a higher PAS rate with deeper insertions, no significant intergroup differences in insertion depth were observed. This suggests that postoperative inflammation and/or a pre-existing degree of angle opening have a greater influence. Future studies should investigate the relationship between angle width and PAS formation.

The IOP at 12 months significantly differs between the two groups based on the PAS status. The PAS-positive group had smaller IOP reductions and higher baseline IOP than the PAS-negative group, although these trends were not statistically significant. Higher preoperative IOP is generally associated with greater IOP reduction after trabeculotomy [6]. A similar trend may also be expected for the iStent inject W, which targets the outflow pathway. However, our findings are inconsistent with this expectation. These results suggest that PAS is associated with adverse postoperative outcomes following iStent inject W and highlight the need for further investigation in larger cohorts.

A TM cleft adjacent to the implanted iStent inject W was observed in 10 stents across 10 eyes. It was significantly more frequent for deep (Grade 2) insertions. The relatively large 360-μm flange of the iStent inject W likely results in greater compressive force on the surrounding TM with deeper placement, predisposing it to cleft formation. However, the frequency of this complication has not been confirmed because no direct comparison with iStent inject has been conducted. Therefore, further investigation is warranted.

Postoperative IOP was significantly lower in eyes with a TM cleft. However, the postoperative medication score was also significantly higher in this group, which likely influenced this outcome. Therefore, the presence of a TM cleft may not independently affect surgical efficacy.

The intraobserver reproducibility for the GS-1–based grading of the implantation site and depth was good, with higher reliability observed for depth. This likely reflects the binary (two-level) depth classification compared with the four-level location grading, as well as the ambiguity of cases where the relatively large flange spans the boundaries between angle structures. Nevertheless, interobserver agreement was not assessed because a single examiner performed grading. Therefore, future studies should include multiple graders to evaluate interobserver reproducibility.

This study had some limitations. First, its retrospective design necessitates confirmation through a prospective study. In particular, the lack of standardized criteria for resuming postoperative topical medications may have influenced outcomes such as the magnitude of IOP reduction. Future studies should employ protocolized postoperative medication strategies. Second, the sample size was modest, rendering the results susceptible to the influence of individual cases. Therefore, validation in larger cohorts is required. Third, insertion depth was graded in two categories based on gonioscopic images; however, heterogeneity within each grade may have obscured depth-related differences in treatment effect. Future studies should employ finer gradations of insertion depth for re-evaluation. Fourth, insertion depth and implant position were analyzed separately despite their likely interaction. Consequently, subsequent studies should evaluate their interactive effects on surgical outcomes. Fifth, the evaluation was conducted solely using findings from automated gonioscopy. The equivalence of these findings with those from manual gonioscopy was not examined. Given that the interpretation of angle configuration can differ between these modalities, this may have influenced the present results. Future studies should assess whether evaluations of insertion depth and implant position of iStent inject W are consistent between automated and manual gonioscopy. Finally, the low statistical power of this study should also be noted. The sample size did not reach the calculated requirement, which may have contributed to the lack of statistically significant differences in the percentage of IOP reduction at 12 months postoperatively among the different insertion-depth groups. Therefore, larger-scale studies with a sufficient sample size are needed in the future.

Despite these limitations, this study has several strengths. To our knowledge, this is the first study to evaluate the insertion depth of the iStent inject W using automated gonioscopy and analyze treatment outcomes in relation to the presence or absence of TM clefts around the device. Our findings suggest that the iStent inject W is less prone to over-insertion than the conventional iStent inject. These findings support those of previous reports [18,19], indicating that gonioscopically visible implants are associated with better postoperative outcomes. Since automated gonioscopy can capture color images of the angle structures, periodic imaging may allow long-term observation of changes in excessively inserted iStent inject W devices, progression of PAS, and potential resolution of trabecular clefts. These advantages indicate the potential for future longitudinal studies.

## 5. Conclusions

The impact of insertion depth on surgical outcomes appears to be minimal in cases where the implantation site of the iStent inject W is confirmed from the GS-1 images. Furthermore, the iStent inject W may present a lower risk of over-insertion.

## Figures and Tables

**Figure 1 jcm-14-07547-f001:**
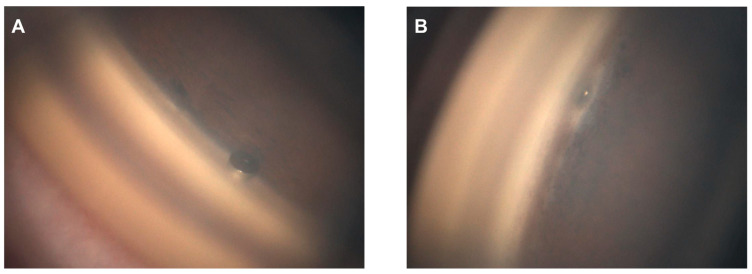
Grading of iStent inject W insertion depth. (**A**) Grade 1—superior edge of the flange protruding into the anterior chamber. (**B**) Grade 2—superior edge of the flange located within the trabecular meshwork.

**Figure 2 jcm-14-07547-f002:**
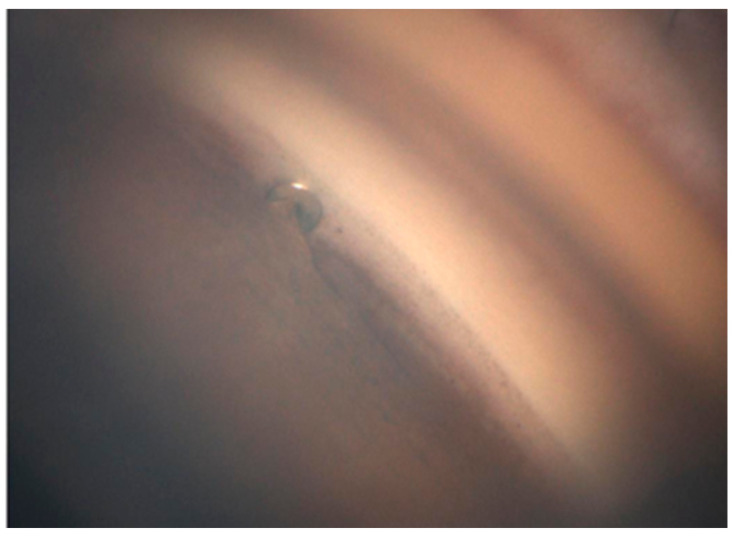
Gonioscopic image showing peripheral anterior synechiae (PAS) around the iStent inject W. PAS is visible on the surface of the iStent inject W flange.

**Figure 3 jcm-14-07547-f003:**
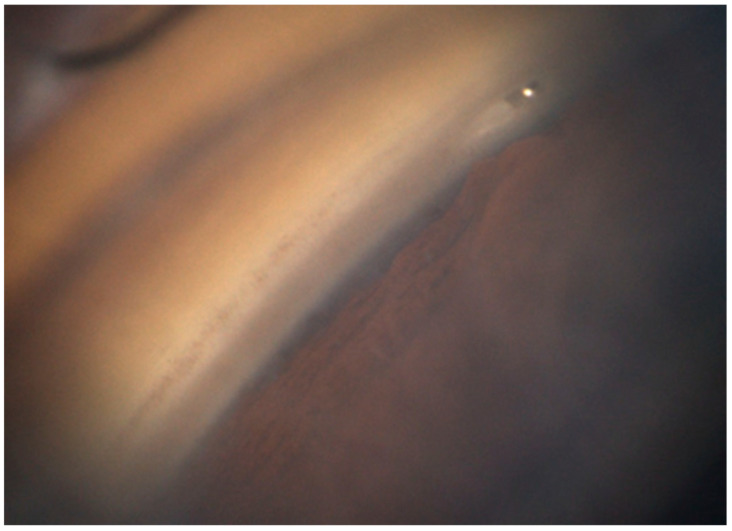
Gonioscopic image showing a trabecular meshwork cleft surrounding the iStent inject W. The device appears to have been over-inserted, as indicated by the trabecular meshwork clefts on both sides.

**Table 1 jcm-14-07547-t001:** Demographic information of patients.

Characteristics	Total
Age, years (range)	68.1 ± 9.7 (50–82)
Sex (Female, eyes, number (%))	Female: 15, 20 eyes (46.5%)
Number of right eyes (%)	21 (51.2%)
Best corrected visual acuity (logMAR) (range)	0.25 ± 0.29 (−0.2 to 1.0)
Spherical equivalent (D) (range)	−4.26 ± 6.08 (−19 to 3.75)
Type of glaucoma	POAG: 38 eyes, PEG: 3 eyes
Preoperative IOP (mmHg) (range)	14.63 ± 4.43 (8.0–33.0)
Preoperative medication score (range)	3.32 ± 1.59 (1.0–6.0)
Mean deviation (dB) (range)	−12.79 ± 8.34 (−30.78 to 1.89)
IOP at 12 months postoperatively (mmHg) (range)	12.05 ± 3.55 (6.0–20.0)
Medication score at 12 months postoperatively (range)	2.44 ± 1.79 (0–5.0)
Position of iStent inject W	Grade 1: 26 (31.7%), Grade 2: 44 (53.7%), Grade 3: 8 (9.8%), Grade 4: 4 (4.9%)
Depth of iStent inject W	Grade 1: 51 (62.2%), Grade 2: 31 (37.8%)

Values are shown as mean ± standard deviation; logMAR: logarithm of the minimum angle of resolution, D: diopter, POAG: primary open-angle glaucoma, PEG: pseudo-exfoliative glaucoma, IOP: intraocular pressure, dB: decibel.

**Table 2 jcm-14-07547-t002:** Comparison by insertion depth.

	Grade 1 + 1	Grade 2 + 2	Grade 1 + 2	Kruskal–Wallis Test
*N*	18	8	15	
Age (range) years	69.9 ± 8.8 (56–82)	63.3 ± 8.3 (50–73)	68.5 ± 11.1 (50–82)	0.351
Type of glaucoma	POAG: 15 eyes PEG: 3 eyes	POAG: 8 eyes	POAG: 15 eyes	0.126 *
Mean deviation (dB)	−12.22 ± 8.37 (−30.78 to 1.89)	−16.78 ± 9.65 (−30.27 to −6.04)	−11.39 ± 7.60 (−22.75 to −0.69)	0.495
Preoperative IOP (mmHg)	14.39 ± 3.09 (10.0–22.0)	12.75 ± 5.09 (8.0–22.0)	15.93 ± 5.24 (11.0–33.0)	0.154
Preoperative medication score	3.44 ± 1.34 (1.0–5.0)	4.13 ± 1.0 (2.0–5.0)	2.73 ± 1.94 (1.0–6.0)	0.144
IOP at 12 months postoperatively (mmHg)	12.39 ± 3.33 (6.0–18.0)	9.50 ± 3.21 (6.0–16.0)	13.0 ± 3.55 (8.0–20.0)	0.044
Medication score at 12 months postoperatively	2.33 ± 1.41 (0–5.0)	3.75 ± 1.83 (0–5.0)	1.87 ± 1.92 (0–5.0)	0.058
IOP reduction rate at 12 months postoperatively (%)	12.81 ± 20.27 (−29.0 to 50.0)	22.85 ± 15.30 (0–45)	16.04 ± 20.41 (−29.0 to 50.0)	0.493

POAG: primary open-angle glaucoma, PEG: pseudo-exfoliative glaucoma, dB: decibel, IOP: intraocular pressure, *: the chi-squared test.

**Table 3 jcm-14-07547-t003:** Comparison by insertion location.

	Group 1	Group 2	Mann–Whitney U Test
*N*	31	10	
Age, years	67.6 ± 10.0	69.7 ± 9.2	0.560
Type of glaucoma	POAG: 28 eyes PEG: 3 eyes	POAG: 10 eyes	0.307 *
Mean deviation (dB)	−12.30 ± 9.08	−14.23 ± 5.74	0.494
Preoperative IOP (mmHg)	15.03 ± 4.91	13.40 ± 2.12	0.300
Preoperative medication score	3.23 ± 1.56	3.60 ± 1.71	0.601
IOP at 12 months postoperatively (mmHg)	11.74 ± 3.88	13.00 ± 2.11	0.099
Medication score at 12 months postoperatively	2.61 ± 1.87	1.90 ± 1.45	0.286
IOP reduction rate at 12 months postoperatively (%)	20.63 ± 17.54	1.44 ± 18.25	0.006

POAG: primary open-angle glaucoma, PEG: pseudo-exfoliative glaucoma, dB: decibel, IOP: intraocular pressure, *: the chi-squared test.

**Table 4 jcm-14-07547-t004:** Comparison based on the presence or absence of a trabecular meshwork cleft (cleft vs. no-cleft).

	Cleft Group	No-Cleft Group	Mann–Whitney U Test
*N*	10	31	
Age, years	63.9 ± 9.4	69.5 ± 9.6	0.143
Type of glaucoma	POAG: 10 eyes	POAG: 28 eyes PEG: 3 eyes	0.307 *
Mean deviation (dB)	−14.75 ± 9.82	−12.13 ± 7.88	0.472
Preoperative IOP (mmHg)	14.40 ± 7.37	14.71 ± 3.13	0.211
Preoperative medication score	3.90 ± 1.85	3.13 ± 1.48	0.138
IOP at 12 months postoperatively (mmHg)	10.40 ± 3.98	12.58 ± 3.29	0.04
Medication score at 12 months postoperatively	3.80 ± 1.81	2.00 ± 1.57	0.0074
IOP reduction rate at 12 months postoperatively (%)	20.76 ± 20.22	13.75 ± 18.91	0.207

POAG: primary open-angle glaucoma, PEG: pseudo-exfoliative glaucoma, dB: decibel, IOP: intraocular pressure, *: the chi-squared test.

**Table 5 jcm-14-07547-t005:** Comparison with and without PAS.

	PAS-Positive Group	PAS-Negative Group	Mann–Whitney U Test
*N*	6	35	
Age, years	66.3 ± 10.2	68.4 ± 9.8	0.627
Type of glaucoma	POAG: 5 eyes PEG: 1 eye	POAG: 33 eyes PEG: 2 eyes	0.341 *
Mean deviation (dB)	−19.19 ± 8.14	−11.75 ± 8.02	0.091
Preoperative IOP (mmHg)	16.83 ± 4.49	14.26 ± 4.37	0.196
Preoperative medication score	4.00 ± 1.10	3.20 ± 1.64	0.376
IOP at 12 months postoperatively (mmHg)	14.83 ± 2.23	11.57 ± 3.53	0.016
Medication score at 12 months postoperatively	3.17 ± 1.17	2.31 ± 1.86	0.284
IOP reduction rate at 12 months postoperatively (%)	8.69 ± 17.41	17.19 ± 19.65	0.284

POAG: primary open-angle glaucoma, PEG: pseudo-exfoliative glaucoma, dB: decibel, IOP: intraocular pressure, PAS: peripheral anterior synechiae, *: the chi-squared test.

## Data Availability

The original contributions presented in this study are included in the article/Appendix A. Further inquiries can be directed to the corresponding author.

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
