# Peer review of "Evaluation of the Insertion Depth of the iStent Inject W and Its Association with Surgical Outcomes Using Automated Gonioscopy"

_jcm, 2025, doi:10.3390/jcm14217547_

Round 1

Reviewer 1 Report

Comments and Suggestions for Authors

This is a retrospective clinical study whose purpose was to evaluate the depth of the iStent inject W using GS-1 automated gonioscope and determine its association with outcomes. A reduced number of studies examine iStent depth evaluation using gonioscopy. However, no study is available in the literature examining the aforementioned parameters. As a result, new studies should definitely contribute contemporary information to this important scientific topic.

This study should address the following:

Comments of major importance:
-The study does not include any power analysis that could determine if the sample size is adequate.

Comments of minor importance:
-Abstract, line 14: Please replace the sentence "To evaluate the depth of the iStent inject W using GS-1 automated gonioscope (GS-1) and determine its association with outcomes." with "The aim of this study is to evaluate the depth of the iStent inject W using GS-1 automated gonioscope (GS-1) and determine its association with outcomes."

-Introduction, line 59: Please replace the phrase "Therefore, we aimed to" with "Therefore, the present study aimed to"

-Table 1:

   -Please add a gap before the parenthesis at the phrase "Female 15, 20 eyes(46.5%)"-> 

"Female 15, 20 eyes (46.5%)".

   -Please add a gap after the colon "POAG:38 eyes"->"POAG: 38 eyes".

   -Please correct the final symbols in the phrase "iStent inject Wの深度"

   -Please replace the phrase "dB(decibel)" with "dB: decibel"

-Table 2: Please add the letter "l" in the "Kruska-Wallis test" -> "Kruskal-Wallis test". Moreover, replace the phrase "dB(decibel)" with "dB: decibel".

-Please replace "Table.3", "Table.4", and "Table.5" with "Table 3.", "Table 4.", and "Table 5."

-Discussion, line 204: Please calculate the percentage of medication score decrease

-Please check the whole manuscript for typographical errors.

In general, the manuscript is interesting, timely, well-written, and the ideas flow logically.

Author Response

Comments 1: -The study does not include any power analysis that could determine if the sample size is adequate.

Response 1: Thank you for this important comment. We agree that statistical power is a crucial issue. Based on this study’s results, the post hoc power calculated was 0.16, indicating a low value. The required sample size was calculated to be significantly larger under this level of power, suggesting that the sample size in this study was insufficient. To address this point, we have added the following sentences to the respective sections in the manuscript:

In the Methods section:

“A power analysis was performed using G*Power (version 3.1.9.7) for the primary endpoint, which is the comparison of the percentage of IOP reduction at 12 months postoperatively among the three depth-defined groups. Additionally, a one-way fixed-effects analysis of variance, as an approximation to the Kruskal–Wallis test, was applied. Statistical power was determined based on the pooled within-group standard deviation and corresponding effect size (Cohen’s f) calculated from our data. Furthermore, the minimum required sample size was estimated using the observed effect size, three groups, a significance level (α) of 0.05, and a statistical power of 0.8.” (pages 4, lines 135–142)

We have also added the following description to the Results section:

“Regarding the analysis of statistical power for the primary endpoint, based on the observed data (means: 12.81%, 22.85%, and 16.04%; standard deviations: 20.27%, 15.30%, and 20.41%; n = 18/8/15), the pooled within-group standard deviation was 19.50, while the corresponding effect size was calculated as Cohen’s f = 0.19. The post hoc power computed from these parameters was 0.16. Ultimately, the required sample size was calculated to be 288 eyes using the estimated effect size, three groups, a significance level (α) of 0.05, and a statistical power of 0.80.” (page 8, lines 207–213)

In addition, we have added the following text to the Discussion section:

“Finally, the low statistical power of this study should also be noted. The sample size did not reach the calculated requirement, which may have contributed to the lack of statistically significant differences in the percentage of IOP reduction at 12 months postoperatively among the different insertion-depth groups. Therefore, larger-scale studies with sufficient sample size are needed in the future.” (page 10, lines 304-309)

Comments of minor importance:

Comments 2: -Abstract, line 14: Please replace the sentence "To evaluate the depth of the iStent inject W using GS-1 automated gonioscope (GS-1) and determine its association with outcomes." with "The aim of this study is to evaluate the depth of the iStent inject W using GS-1 automated gonioscope (GS-1) and determine its association with outcomes."

Response: Thank you for your comment. We have made the necessary correction.

Comments 2: -Introduction, line 59: Please replace the phrase "Therefore, we aimed to" with "Therefore, the present study aimed to"

Response: Thank you for your comment. The correction has been made to the manuscript per your suggestion.

Comments 3: -Table 1:

   -Please add a gap before the parenthesis at the phrase "Female 15, 20 eyes(46.5%)"->

"Female 15, 20 eyes (46.5%)".

   -Please add a gap after the colon "POAG:38 eyes"->"POAG: 38 eyes".

   -Please correct the final symbols in the phrase "iStent inject Wの深度"

   -Please replace the phrase "dB(decibel)" with "dB: decibel"

Response 3: Thank you for your comment. We have made all the above corrections to the manuscript.

Comments 4: -Table 2: Please add the letter "l" in the "Kruska-Wallis test" -> "Kruskal-Wallis test". Moreover, replace the phrase "dB(decibel)" with "dB: decibel".

Response 4: Thank you for your comment. We have made the necessary correction to the manuscript.

Comments 5: -Please replace "Table.3", "Table.4", and "Table.5" with "Table 3.", "Table 4.", and "Table 5."

Response 5: Thank you for your observation. The references to the illustrations have been corrected to “Table 3,” “Table 4,” and “Table 5” as highlighted.

Comments 6: -Discussion, line 204: Please calculate the percentage of medication score decrease

Response 6: Thank you for your comment. The percentage reduction in the medication score was calculated to be 27.33%, and this information has been incorporated into the manuscript accordingly.

Comments 7: -Please check the whole manuscript for typographical errors.

Response 7: Thank you for your comment. We have thoroughly reviewed the manuscript and corrected typographical errors and language issues to improve overall clarity and readability.

Comments 8: In general, the manuscript is interesting, timely, well-written, and the ideas flow logically.

Response 8: Thank you for your positive evaluation and encouraging comments. We sincerely appreciate your thoughtful remarks on the quality and relevance of our work.

Reviewer 2 Report

Comments and Suggestions for Authors

This study describes the association between the depth of an I Stent inject W and the surgical outcome using an automated gonioscopy. 

My main concern is the fact that you only evaluated pictures of automated gonioscopy and did not do a clinical gonioscopy yourself: In my opinion this is insufficient to get a good impression of the chamber angle. 

Another problem of this study is the fact that you did not give enough clinical data about the patients (you state in the discussion that it is a not satisfying to have no controlled medication schedule postoperatively): I would like to know more clinical data about the patients, because the type of glaucoma and the stage of glaucoma also influence the postoperative outcome. Please give a detailed report about this issue.

In the introduction you talk in one phrase about trabeculectomy and trabeculotomy - these are two totally different operations and you can not compare them the way you did - please rewrite this part of the introduction.

In my opinion the study has not much clinical significance and this needs to be mentioned in the discussion.

Comments on the Quality of English Language

Some phrases are difficult to understand, for example "The influence of the depth on the outcomes appears to be limited in eyes for which the iStent inject W was visualized using GS-1". This phrase could be ameliorated. As there are other phrases which are difficult to understand, I recommend that an English native speaker works on the manuscript.

Author Response

Comment1: My main concern is the fact that you only evaluated pictures of automated gonioscopy and did not do a clinical gonioscopy yourself: In my opinion this is insufficient to get a good impression of the chamber angle. 

Response 1: Thank you for your thoughtful and insightful comment. We agree that ideally, manual gonioscopy should have been performed and its findings analyzed in parallel. However, automated gonioscopy has the advantage of allowing clear-angle images to be obtained easily and within a short period of time. Automated gonioscopy is essentially performed under bright illumination; therefore, the assessment of angle closure may differ from that of manual gonioscopy. Nevertheless, when evaluating the position and depth of the iStent inject W, these factors are unlikely to be influenced by lighting conditions. Therefore, we believe there would be a slight difference between automated and manual gonioscopy in this context. In the present study, we based our evaluations solely on the findings of automated gonioscopy.

As you pointed out, we did not examine whether the findings from manual and automated gonioscopy are truly equivalent, and we acknowledge this as a limitation of our research that should be addressed in future studies. We have added the following statement to the Discussion section:

“Fifth, the evaluation was conducted solely using findings from automated gonioscopy. The equivalence of these findings with those from manual gonioscopy was not examined. Given that the interpretation of angle configuration can differ between these modalities, this may have influenced the present results. Future studies should assess whether evaluations of insertion depth and implant position of iStent inject W are consistent between automated and manual gonioscopy.” (pages  10, lines 299- 304)

Comments 2: Another problem of this study is the fact that you did not give enough clinical data about the patients (you state in the discussion that it is a not satisfying to have no controlled medication schedule postoperatively): I would like to know more clinical data about the patients, because the type of glaucoma and the stage of glaucoma also influence the postoperative outcome. Please give a detailed report about this issue.

Response 2: Thank you for your valuable comment. We agree that confirming potential differences in clinical characteristics among the subgroups in each analysis is important. Therefore, we have added age, glaucoma type, and mean deviation (MD) to each table. No significant differences were found in MD or glaucoma type among the three groups. The revised tables are shown below.

Table 2. Comparison by insertion depth

Grade 1+1

Grade 2+2

Grade 1+2

Kruskal–Wallis test

N

18

8

15

Age

(range) years

69.9 ± 8.8

(56–82)

63.3 ± 8.3

(50–73)

68.5 ± 11.1

(50–82)

0.351

Type of glaucoma

POAG: 15 eyes

PEG: 3 eyes

POAG: 8 eyes

POAG: 15 eyes

0.126*

Mean deviation (dB)

-12.22±8.37

(-30.78 to 1.89)

-16.78±9.65

(-30.27 to -6.04)

-11.39±7.60

(-22.75 to -0.69)

0.495

Preoperative IOP (mmHg)

14.39±3.09

(10.0–22.0)

12.75±5.09

(8.0–22.0)

15.93±5.24

(11.0–33.0)

0.154

Preoperative medication score

3.44±1.34

(1.0–5.0)

4.13±1.0

(2.0–5.0)

2.73±1.94

(1.0–6.0)

0.144

IOP at 12 months postoperatively (mmHg)

12.39±3.33

(6.0–18.0)

9.50±3.21

(6.0–16.0)

13.0±3.55

(8.0–20.0)

0.044

Medication score at 12 months postoperatively

2.33±1.41

(0–5.0)

3.75±1.83

(0–5.0)

1.87±1.92

(0–5.0)

0.058

IOP reduction rate at 12 months postoperatively (%)

12.81±20.27

(-29.0 to 50.0)

22.85±15.30

(0–45)

16.04±20.41

(-29.0 to 50.0)

0.493

POAG: primary open-angle glaucoma, PEG: pseudo-exfoliative glaucoma, dB: decibel, IOP: intraocular pressure, *: the chi-squared test

Table 3. Comparison by insertion location

Group 1

Group 2

Mann–Whitney U test

N

31

10

Age, years

67.6±10.0

69.7±9.2

0.560

Type of glaucoma

POAG: 28 eyes

PEG: 3 eyes

POAG: 10 eyes

0.307*

Mean deviation (dB)

-12.30±9.08

-14.23±5.74

0.494

Preoperative IOP (mmHg)

15.03±4.91

13.40±2.12

0.300

Preoperative medication score

3.23±1.56

3.60±1.71

0.601

IOP at 12 months postoperatively (mmHg)

11.74±3.88

13.00±2.11

0.099

Medication score at 12 months postoperatively

2.61±1.87

1.90±1.45

0.286

IOP reduction rate at 12 months postoperatively (%)

20.63±17.54

1.44±18.25

0.006

POAG: primary open-angle glaucoma, PEG: pseudo-exfoliative glaucoma, dB: decibel, IOP: intraocular pressure, *: the chi-squared test

Table 4. Comparison based on the presence or absence of a trabecular meshwork cleft (cleft vs. no-cleft)

Cleft group

No-cleft group

Mann–Whitney U test

N

10

31

Age, years

63.9±9.4

69.5±9.6

0.143

Type of glaucoma

POAG: 10 eyes

POAG: 28 eyes

PEG: 3 eyes

0.307*

Mean deviation (dB)

-14.75±9.82

-12.13±7.88

0.472

Preoperative IOP (mmHg)

14.40±7.37

14.71±3.13

0.211

Preoperative medication score

3.90±1.85

3.13±1.48

0.138

IOP at 12 months postoperatively (mmHg)

10.40±3.98

12.58±3.29

0.04

Medication score at 12 months postoperatively

3.80±1.81

2.00±1.57

0.0074

IOP reduction rate at 12 months postoperatively (%)

20.76±20.22

13.75±18.91

0.207

POAG: primary open-angle glaucoma, PEG: pseudo-exfoliative glaucoma, dB: decibel, IOP: intraocular pressure, *: the chi-squared test

Table 5. Comparison with and without PAS

PAS-positive group

PAS-negative group

Mann–Whitney U test

N

6

35

Age, years

66.3±10.2

68.4±9.8

0.627

Type of Glaucoma

POAG: 5 eyes

PEG: 1 eye

POAG: 33 eyes

PEG: 2 eyes

0.341*

Mean deviation (dB)

-19.19±8.14

-11.75±8.02

0.091

Preoperative IOP (mmHg)

16.83±4.49

14.26±4.37

0.196

Preoperative medication score

4.00±1.10

3.20±1.64

0.376

IOP at 12 months postoperatively (mmHg)

14.83±2.23

11.57±3.53

0.016

Medication score at 12 months postoperatively

3.17±1.17

2.31±1.86

0.284

IOP reduction rate at 12 months postoperatively (%)

8.69±17.41

17.19±19.65

0.284

POAG: primary open-angle glaucoma, PEG: pseudo-exfoliative glaucoma, dB: decibel, IOP: intraocular pressure, PAS: peripheral anterior synechiae, *: the chi-squared test

Comments 3: In the introduction you talk in one phrase about trabeculectomy and trabeculotomy - these are two totally different operations and you can not compare them the way you did - please rewrite this part of the introduction.

Response 3: Thank you for your insightful comment. We fully agree with your observation and divided the text by surgical approach and revised it as follows:

“Conventional surgical techniques for glaucoma are broadly categorized into filtration procedures (such as trabeculectomy) and trabeculotomy, which targets the trabecular outflow pathway. Although trabeculectomy achieves substantial IOP reduction, it requires conjunctival dissection and iridectomy, making it relatively invasive. Traditional trabeculotomy was also performed ab externo, similarly necessitating a conjunctival incision and potentially compromising subsequent trabeculectomy in some cases.” (page 1, lines 36–41)

Comments 4: In my opinion the study has not much clinical significance and this needs to be mentioned in the discussion.

Response 4: Thank you for your valuable comment. As you pointed out, we agree that the small sample size and low statistical power are key limitations of this study. However, we believe that our work retains clinical and scientific significance in several respects. Specifically, this study newly evaluated the insertion depth of the iStent inject W using automated gonioscopy and examined its relationship with surgical outcomes. Furthermore, we also investigated and reported, for the first time to our knowledge, cases where a trabecular meshwork cleft was observed around the iStent inject W and analyzed whether its presence affected treatment results.

Our findings suggest that the iStent inject W is less prone to over-insertion compared with the conventional iStent inject. Furthermore, our results also support previous reports indicating that cases where the implant can be visualized gonioscopically tend to have favorable postoperative outcomes. Compared with anterior segment OCT, automated gonioscopy provides color images of the angle structures, facilitating longitudinal assessment of how excessively inserted iStent inject W devices change over time, whether peripheral anterior synechiae (PAS) increase in cases where they are present, and whether trabecular clefts resolve in such eyes. These aspects highlight the potential of automated gonioscopy for future longitudinal research.

To address this point, we have added the following sentences to the Discussion section:

“Finally, the low statistical power of this study should also be noted. The sample size did not reach the calculated requirement, which may have contributed to the lack of statistically significant differences in the percentage of IOP reduction at 12 months postoperatively among the different insertion-depth groups. Therefore, larger-scale studies with sufficient sample size are needed in the future.

Despite these limitations, this study has several strengths. To our knowledge, this is the first study to evaluate the insertion depth of the iStent inject W using automated gonioscopy and analyze treatment outcomes in relation to the presence or absence of TM clefts around the device. Our findings suggest that the iStent inject W is less prone to over-insertion than the conventional iStent inject. These findings support those of previous reports [18,19], indicating that gonioscopically visible implants are associated with better postoperative outcomes. Since automated gonioscopy can capture color images of the angle structures, periodic imaging may allow long-term observation of changes in excessively inserted iStent inject W devices, progression of PAS, and potential resolution of trabecular clefts. These advantages indicate the potential for future longitudinal studies.” (pages 10–11, lines 304–317)

Comments 5: Some phrases are difficult to understand, for example "The influence of the depth on the outcomes appears to be limited in eyes for which the iStent inject W was visualized using GS-1". This phrase could be ameliorated. As there are other phrases which are difficult to understand, I recommend that an English native speaker works on the manuscript.

Response 5: Thank you for your comment. We acknowledge the need for improved clarity in some parts of the manuscript. Accordingly, the manuscript has been thoroughly reviewed and revised by a native English speaker to enhance the readability and precision of the language. The suggested sentence, along with other unclear phrases, has been reworded, and the corrections have been implemented throughout the text.

Round 2

Reviewer 2 Report

Comments and Suggestions for Authors

Thank you, you addressed my comments and find the changed manuscript ready for publication from my side

Comments on the Quality of English Language

Some phrases are difficult to understand, for example "The influence of the depth on the outcomes appears to be limited in eyes for which the iStent inject W was visualized using GS-1". This phrase could be ameliorated. As there are other phrases which are difficult to understand, I recommend that an English native speaker works on the manuscript.